# Role of syndecan-1 in the interaction between dendritic cells and T cells

M. Kouwenberg[1], A. Rops[1], M. Bakker-van Bebber[1], L. Diepeveen[1], M. Götte [2], L. Hilbrands[1], J. van der Vlag [1]*

1 Department of Nephrology, Radboud University Medical Center, Nijmegen, The Netherlands,
2 Department of Gynecology and Obstetrics, University of Münster, Münster, Germany

* johan.vandervlag@radboudumc.nl

**Data Availability Statement:** All relevant data are within the manuscript and its Supporting Information files.

## Abstract

Syndecan-1 (Sdc-1) is a heparan sulfate proteoglycan that can bind cytokines and chemokines via its heparan sulfate side chains, and has immunomodulatory properties in experimental models. Sdc-1 expression has been reported on dendritic cells (DC) and T cells. The potential role of Sdc-1 in DC—T cell interaction has not been investigated yet. We postulate that Sdc-1 is involved in DC–T cell interaction and may influence graft survival in an allogeneic transplant model. Sdc-1 expression on bone marrow-derived DC and T cells was analyzed by flow cytometry. Unstimulated and LPS stimulated Sdc-1 deficient DC were evaluated *in vitro* for phenotype and stimulatory capacity in mixed lymphocyte reaction. Sdc-1 deficient T cells were evaluated for proliferative capacity and differentiation in a mixed lymphocyte reaction and a proliferation assay. Allograft survival was evaluated in a fully MHC mismatched heterotopic heart transplant model, with either Sdc-1 deficient donors or recipients. Sdc-1 was expressed on the cell surface of unstimulated and LPS matured DC. Sdc-1 deficiency had no effect on expression of co-stimulatory molecules, cytokine production or T cell stimulatory capacity as compared to WT DC. Sdc-1 expression was not detectable on WT T cells, although intracellular Sdc-1 expression could be demonstrated after ConA activation. Sdc-1 deficient T cells showed reduced proliferation upon DC or ConA stimulation and reduced IL-17 production upon ConA stimulation, compared to WT T cells. Sdc-1 deficiency of either allograft or recipient did not prolong allograft survival. In conclusion, Sdc-1 is expressed on the cell surface of DC, where its absence does not affect DC phenotype or T cell stimulatory capacity. Sdc-1 is intracellularly expressed in ConA activated T cells. Sdc-1 deficiency in T cells results in a reduced proliferative response *in vitro*, as induced by DC and ConA. Sdc-1 deficiency in donor or recipient does not affect allograft survival.

## Introduction

Dendritic cells (DC) are antigen presenting cells that contribute to host defense by endocytosis of pathogens and presentation of the processed pathogen to a naive T cell in secondary lymphoid organs. T cells are activated by three DC-derived signals: 1. presentation of the

**Funding:** JvdV; LH; MK: NWO ZonMw AGIKO 92003567, Radboud PhD programm 2010. The funders had no role in study design, data collection and analysis, decision to publish, or preparation of the manuscript.

**Competing interests:** The authors have declared that no competing interests exist.

processed pathogen as a peptide in an MHC molecule; 2. stimulation by cell surface co-stimulatory molecules; and 3. cytokines. Based on the combination of these three signals the naive T cell will differentiate into a particular subset of T helper cells that further direct the immune response. Apart from these well documented signals, heparan sulfate proteoglycans (HSPGs) can also affect the interaction between DC and T cells. Agrin, neuropilin-1, and syndecan-4 [1–4] have been shown to be involved in DC–T cell interaction and contribute to T cell activation. Whereas DC and T cells also express several other HSPGs, the functional role of these HSPGs in DC and T cells, as well as their potential role in DC–T cell interaction is largely unknown.

One of these HPSGs is Sdc-1 or CD138. It is mainly known by its expression on plasma cells and various types of epithelial cells and is involved in cell proliferation, cell growth, metastasis, and angiogenesis. The core protein of Sdc-1 consists of an extracellular, a transmembrane, and a cytoplasmatic domain. The extracellular part–or ectodomain–is decorated by chondroitin sulfate and at least one heparan sulfate chain (reviewed in [5, 6]. Heparan sulfate (HS) consists of repeating disaccharide units of an N-acetylglucosamine and glucuronic acid. HS elongation and modification, i.e. C5 epimerization and sulfation at multiple positions, is regulated by about 30 enzymes. HS is highly heterogenous in structure due to distinct sulfation patterns which results in specific binding of a myriad of factors including cytokines and chemokines [7]. The binding of cytokines and chemokines via the HS side chain has multiple functions: 1. Protection against proteolytic degradation; 2. Functioning as a co-receptor; 3. Establishing a local concentration gradient. In an immunological context HS is a key player in binding of CXCL1, thereby regulating neutrophil influx [8], and increasing efficiency of CXCR2 ligation [9]. Given the fact that HS chains serve different functions in binding cytokines, it is not surprising that interference in this glycosaminoglycan—cytokine interaction can alter various immunological responses [10]. In mice, Sdc-1 deficiency increases the number of interleukin (IL)-17 producing gamma-delta T cells, leading to exacerbated skin inflammation in experimentally induced psoriasisform dermatitis [11]. In other mouse models of inflammation, Sdc-1 deficiency results in exaggerated airway hyperresponsiveness [12], increased multi-organ injury in endotoxemia and toxic shock [13], enhanced disease severity in experimental autoimmune encephalitis [14], increased renal damage due to ischemia-reperfusion [15], increased T cell driven inflammation in an abdominal aortic aneurysm model [16], and exacerbated glomerulonephritis [17]. In addition, a role for syndecan-1 in polarization of T helper cell subsets has been demonstrated in human breast cancer tissue [18].These studies suggest that Sdc-1 has an immunosuppressive and anti-inflammatory role. After shedding from the cell surface, Sdc-1 can also exert different functions as soluble HSPG. Shedded Sdc-1 can activate neutrophils [19] and *in vivo* it has been shown to reduce neutrophil-mediated inflammation by neutralization of sequestered CXCL1 [20], which could also explain why inflammatory conditions are more aggravated in Sdc-1 deficient mouse models as outlined above.

While the immunomodulatory properties of Sdc-1 have been established in mouse models of inflammation, there is little data on the potential role of Sdc-1 in transplantation. In kidney transplant patients and animal models, increased tubular Sdc-1 expression was suggested to promote tubular survival and repair, while increased Sdc-1 plasma levels reflected early loss of tubular function [15, 21]. The effect of Sdc-1 deficiency on allograft survival was not investigated.

In mice, Sdc-1 expression has been described on plasma cells, DC, M2 macrophages, IL-17 producing gamma-delta T cells, and the NKT17 subset of invariant natural killer T (NKT) cells [11, 22, 23], and intracellular expression was reported for CD4+ T cells [4]. Sdc-1 has been reported to affect macrophage motility as well as macrophage polarization towards the more

immunoregulatory M2 phenotype [22]. In line with the effect on macrophage motility, Sdc-1 was shown to affect DC migration while no effect on DC maturation and DC-mediated T cell activation was observed [24].

Sdc-1 was suggested to affect T cell functioning in a mouse model of gram positive septic shock [13]. Sdc-1 deficient mice showed reduced survival and increased systemic cytokine levels upon Staphylococcal enterotoxin B-induced septic shock compared to wild-type mice. Depletion of T cells protected the mice against the effects caused by Sdc-1 deficiency.

We hypothesized that Sdc-1 is involved in DC–T cell interaction, with Sdc-1 deficiency potentially resulting in an unrestrained T cell response upon DC stimulation. We examined this in *in vitro* experiments with DC and T cells obtained from Sdc-1 deficient mice. To evaluate the role of Sdc-1 in graft rejection, we used a heart transplantation model in mice with Sdc-1 deficiency in either the donor or the recipient.

## Material and methods

### Mice

Male mice C57Bl/6 (H-2$^d$), Balb/c (H-2$^b$) (Charles River laboratories, USA) and male Sdc-1 knockout mice on a C57Bl/6 background [25] were housed under specified pathogen-free conditions. Sdc-1 knockout mice were genotyped with gene specific primers as described previously [26]. All animal experiments were carried out after permission granted by the animal ethics committee of the Radboud University Nijmegen (Permit Number 2011–024). Animals were handled according to the guidelines of the local animal welfare body of the Radboud University Nijmegen.

### Dendritic cell culture

DC were cultured from bone marrow obtained from wild-type (WT) C57Bl6, Sdc-1$^{-/-}$ and Balb/c mice according to a method adopted from Lutz et al. [27, 28]. In short, femora and tibiae were harvested after cervical dislocation and bone marrow was flushed with medium (RPMI-1640 Dutch modification (Invitrogen, USA), supplemented with 50 μM β-mercaptoethanol (Sigma-Aldrich, St Louise, USA), 1% glutamax (Invitrogen), 1% pyruvate (Invitrogen) and 10,000 U/ml penicillin-streptomycin (Invitrogen). Cells were suspended in medium with 10% fetal calf serum (BioWhittaker, Lonza, Basel, Switzerland), supplemented with 20 ng/ml rGM-CSF (PeproTech, Rocky Hill, USA) and subsequently cultured for 9 days in six well plates (0.8 x 10$^6$ cells/well, Corning Incorporated, USA) at 37°C and 5% CO$_2$. At day 3 and day 6 medium was refreshed. After 8 days of culture, DC were stimulated with different TLR ligands for 24 hours, i.e for TLR2 stimulation 1 μg/ml PAM$_3$CysSerLys$_4$ (PAM$_3$, tlrl-pms, InvivoGen), for TLR4 stimulation 1 μg/ml LPS Ultra pure (InvivoGen, San Diego, USA), and for TLR9 stimulation 5 μg/ml ODN1826 (tlrl-1826, InvivoGen). Unstimulated DC received no additives. At day nine, cells were harvested. Culture supernatant was collected and stored at -20°C for cytokine measurement. Cells were analyzed by flow cytometry or used in co-culture with T cells.

### Mixed lymphocyte reaction, Concanavalin A induced cell proliferation and apoptosis assays

The capacity of Sdc-1 deficient DC to activate Balb/c T cells and the capacity of Sdc-1 deficient T cells to proliferate upon stimulation by Balb/c DC were tested in mixed lymphocyte reactions (MLR). For T cell enrichment of splenocytes, a negative selection with MHCII microbeads and a LS column (Miltenyi Biotec GmbH, Gladbach, Germany) was performed. Before

enrichment, the splenocytes contained 38% T cells and 50% B cells, which changed to 82% T cells and 5% B cells after enrichment. The response of Sdc-1 deficient splenocytes after exposure to Concanavalin A (ConA) was tested in a proliferation assay. ConA is a known T cell mitogen that does not induce B cell proliferation [29, 30]. Splenocytes were harvested from the spleen of Balb/c, C57Bl6 or Sdc-1$^{-/-}$ mice and erythrocytes were lysed by short incubation in Ammonium-Chloride-Potassium lysis buffer. To analyze proliferation, we used an intracellular staining of splenocytes/T cells with CFDA-SE (Molecular Probes, Life Technologies Ltd, Paisley, UK) according to the manufacturer's instructions. For MLR, 1 x 10$^5$ CFSE labeled T cells were resuspended in medium with 10% fetal calf serum and co-cultured with 2.5 x 10$^4$ DC in a 96 wells round bottom plate (Costar, Corning Inc. USA) for 4 to 6 days at 37˚C and 5% $CO_2$. For proliferation assay, 1 x 10$^5$ CFSE labeled cells were resuspended in medium with 10% fetal calf serum and cultured in a 96 wells round bottom plate (Costar) for 4 to 6 days with different concentrations of ConA (Sigma) at 37˚C and 5% $CO_2$. Proliferation of cells was analyzed by dilution of the CFSE signal using flow cytometry. Apoptosis of 5 x 10$^5$ cells/ml splenocytes was induced by incubation with 0.6 to 120 ng/ml 4-nitroquinoline 1-oxide (4-NQO, Sigma) for 18 hours at 37˚C and 5% $CO_2$ as we described previously [28], and the degree of apoptosis was analyzed by annexin-V and propidium iodide staining using flow cytometry.

## Flow cytometry

Antibodies were diluted in PBS supplemented with 0.5% bovine serum albumin (BSA, Sigma). To analyze DC maturation cells were stained with anti-CD11c (clone N418, IQ products, Groningen, the Netherlands), anti-CD40 (clone FGK45.5, Miltenyi Biotec GmbH, Gladbach, Germany), anti-CD80 (clone 16-10A1, Biolegend, Fell, Germany), anti-CD86 (clone PO3.1, eBioscience, San Diego, USA), and anti-MHCII (clone M5/114.15.2, eBioscience, Vienna, Austria). Cells were gated for CD11c+ cells and subsequently analyzed for expression of CD40, CD80, CD86 or MHCII. Purity of MHCII-depleted splenocytes was analyzed by anti-CD3 (Clone 145-2C11, BD Biosciences, New Jersey, USA) and anti-CD19 (Clone 1D3, BD Biosciences) reactivity of both fractions. To analyze splenocyte composition cells were stained with anti-CD3 (Clone 145-2C11, BD Biosciences), anti-CD4 (Clone H129.19, BD Biosciences), anti-CD8 (Clone 53–6, BD Biosciences), and anti-CD19 (Clone 1D3, BD Biosciences). T cells and DC were stained with anti-syndecan-1 (rat IgG$_{2A}$κ monoclonal antibody, clone 281–2, BD Biosciences) or isotype control (rat IgG$_{2A}$κ, BD Biosciences) and conjugate (Alexa488, goat anti rat, Molecular Probes, Eugene, USA) to analyze extracellular Sdc-1 expression. To reduce non-specific binding, secondary antibody was diluted in PBS with 0.5% BSA and 10% normal goat serum. For analysis of intracellular Sdc-1 expression, cells were fixed and permeabilized (Fix&Perm kit, eBioscience) and stained with anti-Sdc-1 or isotype. For permeabilization, antibodies were diluted in permeabilization buffer (eBioscience). For annexin-V and propidium iodide staining, an Annexin-V PI staining kit (BioVision, Milpitas, USA) was used according to the manufacturer's instruction. Cells were analyzed by flow cytometry (FC 500, Beckman Coulter, Brea, USA) and data were analyzed with the CFX software package (Beckman Coulter).

## ELISA

In DC culture supernatant, MLR and proliferation assay supernatant and plasma of transplanted mice The following cytokines and chemokine were measured in supernatants of DC culture, MLR and proliferation assays, and in mouse plasma: IL-2 (, IL-4, IL-6, IL-12, IL-17,

IL-21, IL-23, interferon-γ, TNF-α (all eBioscience) and IL-10 and CXCL1 (R&D systems, Minneapolis, USA) according to the manufacturer's instructions.

## Heterotopic heart transplant model

Heterotopic (intra-abdominal) vascularized heart transplantation was performed using a method described by Corry et al. [31] and previously performed in our group [32, 33]. Balb/c mice served as donor for either WT or Sdc-1$^{-/-}$ mice, and hearts from WT or Sdc-1$^{-/-}$ mice were transplanted in Balb/c mice. Recipient mice were anesthetized by isoflurane inhalation and subcutaneous injection of FFM-mix (fentanylcitrate, fluanisone, midazolam). Donor mice were administered FFM mix prior to surgery. After surgery, intramuscular injections of buprenorfine were used as analgesic medication. Mice were daily monitored for weight and wellbeing (general appearance and behavior, appearance of fur and mucous membranes, respiration rate, and signs of dehydration). Graft survival was monitored by daily transabdominal palpation of the donor heart. Rejection was scored when there was a strong decrease in strength or absence of heart pulsations. Upon rejection, blood samples were obtained from the retro-orbital sinus under isoflurane anesthesia. Blood samples were collected in ethylenediaminetetraacetic acid tubes (BD Biosciences) and immediately stored on ice. Tubes were spinned 3500 x $g$ for 10 minutes at 4˚C and plasma was stored at -80˚C. After blood collection, mice were sacrificed by cervical dislocation.

## Statistical analysis

Statistical analysis was performed with GraphPad Prism (version 5.0 for Windows, GraphPad Software, San Diego, USA). Results are expressed as mean ± SEM. For comparisons between groups we used Mann-Whitney test and unpaired t-test where appropriate. Graft survival was compared using Kaplan-Meier curves and the log rank test. A P-value ≤ 0.05 was considered significant.

## Results

### Maturation-induced Sdc-1 expression on bone marrow derived DC does not affect DC phenotype or stimulatory capacity

Unstimulated DC express a low level of Sdc-1 at their cell surface (Fig 1A, left panel). In addition, after permeabilization of the cells (Fig 1A, right panel), we found no evidence for additional intracellular expression. Upon (LPS induced) maturation of the DC, extracellular Sdc-1 expression by DC increased slightly (Fig 1A, left panel), while intracellular expression was unaffected (Fig 1A, right panel).

To evaluate the effect of Sdc-1 deficiency on expression of co-stimulatory molecules and MHCII after DC maturation, WT and Sdc-1 deficient DC were incubated with 1 μg/ml ultrapure LPS (TLR4 agonist) during the final 24 hours of culture. After 9 days of culture, CD11c + DC were evaluated for the expression of CD40, CD80, CD86, and MHCII, and culture supernatants were analyzed for cytokine production. The LPS-induced increase in expression of co-stimulatory molecules and MHCII on CD11c+ DC was similar in Sdc-1$^{-/-}$ and WT DC (Fig 1B). As expected, stimulation with LPS resulted in a strong increase in the production of pro-inflammatory cytokines by DC, but also here there was no difference between WT and Sdc-1 deficient DC (Fig 1C).

Based on the participation of Sdc-1 in binding different types of pathogens [34–37], we hypothesized that Sdc-1 could be involved in TLR2, and TLR9 mediated DC maturation. Therefore, Sdc-1 deficient and WT DC were incubated during the last 24 hours of culture with

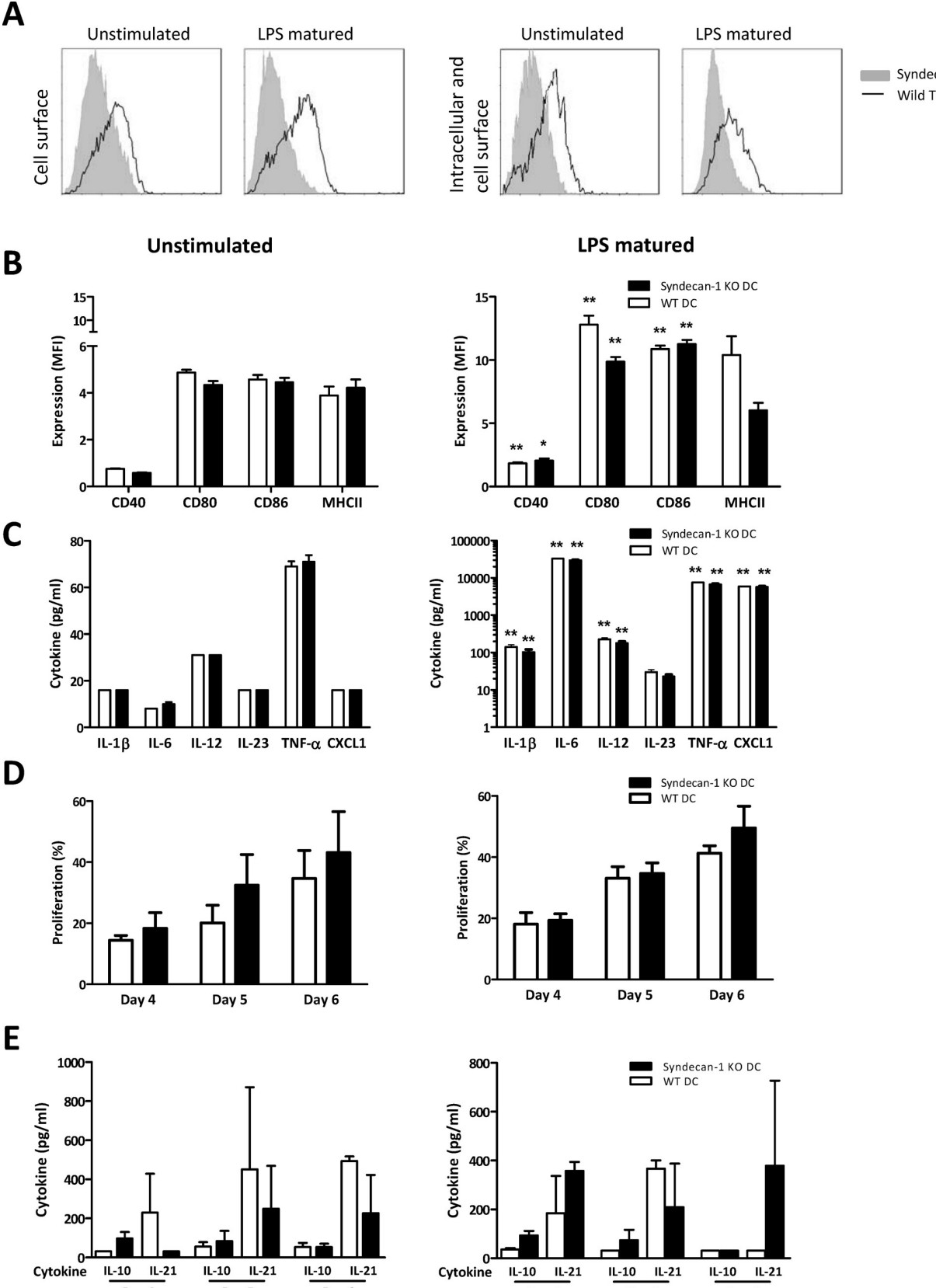

**Fig 1. Sdc-1 deficiency does not affect DC maturation or function.** Sdc-1 expression at the cell surface (A, left panel) or intracellularly and at the cell surface (A, right panel) by unstimulated and LPS matured WT (black line) and Sdc-1-deficient (grey area) DC as measured by flow cytometry (A). Expression of co-stimulatory molecules and MHCII by unstimulated (B, left panel) or LPS matured (B, right panel) WT and Sdc-1-deficient CD11c+ DC as measured by flow cytometry (B). Cytokine profile of unstimulated (C, left panel) or LPS matured (C, right panel) WT and Sdc-1-deficient DC as measured by ELISA. The increase in expression of co-stimulatory molecules and in DC derived cytokine and CXCL1 production upon LPS exposure is significant compared to unstimulated DC. T cell stimulatory capacity of unstimulated (D, left panel) or LPS matured (D, right panel) WT and Sdc-1-deficient DC as measured by CFSE dilution using flow cytometry. T cell derived cytokine production in co-cultures of T cell and unstimulated (E, left panel) or LPS matured (E, right panel) WT and Sdc-1-deficient DC as measured by ELISA. Levels of IL-4 and TNF-α were undetectable. All experiments were replicated 3–5 times. Results are expressed as means ± standard error of means. * p< 0.05, ** p< 0.01 for comparison between unstimulated and LPS matured DC; Mann Whitney test.

1 μg/ml PAM$_3$CysSerLys$_4$ (TLR2 agonist) or 5 μg/ml ODN1826 (TLR9 agonist). Stimulation with ODN1826 resulted in decreased CD80 expression in Sdc-1 deficient DC as compared to WT DC. Sdc-1 deficiency did not alter expression of other co-stimulatory molecules and MHCII, or cytokine production (S1 Fig).

Although Sdc-1 deficiency affected TLR mediated DC maturation only mildly, Sdc-1 could be involved in activation of T cells and thereby influence the adaptive immune response. Therefore, we evaluated the role of Sdc-1 on DC in T cell activation, proliferation and differentiation in an allogeneic MLR. Neither T cell proliferation nor co-culture cytokine profile was altered by deficiency of Sdc-1 in the DC, as shown in Fig 1D and 1E, respectively. The stimulatory capacity of TLR2 or TLR9 ligand matured DC was not affected by Sdc-1 deficiency either (S1 Fig).

We conclude that DC cell surface Sdc-1 expression increases upon (LPS induced) maturation, while Sdc-1 deficiency has no effect on either DC phenotype or the capacity of DC to stimulate T cells.

## Sdc-1 deficiency in T cells results in reduced DC-induced T cell proliferation

Although Sdc-1 deficiency does not affect the stimulatory capacity of DC, we speculated that Sdc-1 could play a role in T cell functioning upon DC stimulation. Cell surface expression of Sdc-1 has been described for human T cells [38, 39] and mouse IL-17 producing NKT cells and gamma-delta T cells[11, 23], but not for mouse splenic T cells [4]. In line with this, we found no Sdc-1 expression on the cell surface of unstimulated splenic T cells (Fig 2A, left panel). After stimulation with ConA, we found a weak intracellular Sdc-1 expression in WT splenic T cells (Fig 2A, right panel).

To further evaluate the role of Sdc-1 in T cells, we performed MLR with unstimulated or LPS stimulated Balb/c DC as stimulator cells and Sdc-1 deficient or WT T cells as responder cells. Sdc-1 deficient T cells showed a reduced proliferative response upon stimulation by unstimulated DC compared to WT T cells (Fig 2B, left panel). However, the proliferative response of Sdc-1 and WT T cells was similar when co-cultured with LPS matured DC (Fig 2B, right panel).

Next, we analyzed several cytokines in supernatants of co-cultures of DC and T cells. Upon stimulation with either immature DC or LPS-matured DC, levels of IL-17, INF-y and TNF-α were comparable (Fig 2C, 2D and 2E).

## Sdc-1 deficiency in splenocytes reduces ConA-induced proliferation

Since we found a reduced proliferative response of Sdc-1 deficient T cells in co-cultures with (unstimulated) DC compared to WT T cells, we wanted to investigate whether this effect was specific for DC stimulation. Therefore, we cultured splenocytes in the presence of the T cell

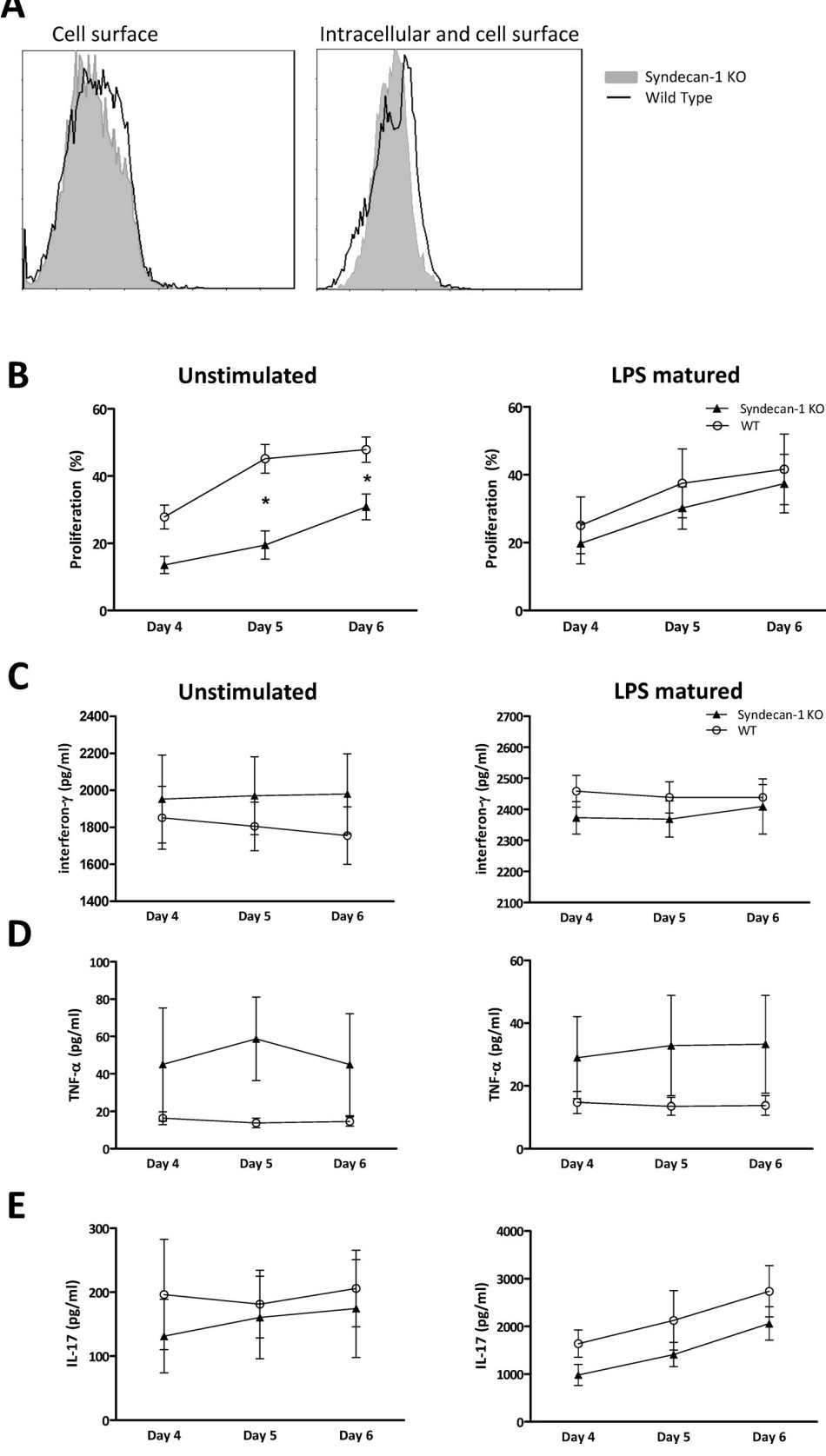

**Fig 2. Sdc-1 deficient T cells display a reduced proliferative response in co-culture with DC.** Sdc-1 expression at the cell membrane of naïve WT (A, left panel, black line) and Sdc-1-deficient splenic T cells (A, left panel, grey area), and intracellular and cell surface expression of ConA (0.25 μg/ml) -activated WT (A, right panel, black line) and Sdc-1 deficient T cells (A, right panel, grey area). T cell proliferation in co-cultures of WT and Sdc-1 deficient T cells with unstimulated DC (B, left panel) or LPS matured DC (B, right panel) as analyzed by CFSE dilution using flow cytometry. IFN-γ (C), TNF-α (D) and IL-17 (E) production in co-cultures of WT and Sdc-1 deficient T cells with either unstimulated DC (C, D, E, left panels) or LPS matured DC (C, D, E, right panels) as measured by ELISA. Control conditions with T cells only showed only minimal proliferation (max. 2–4% at day 6). Expression of proliferation and cytokine levels reflect means and standard error of means of 4 independent experiments. Mann Whitney test, * p< 0.05 for comparison between WT and Sdc-1 deficient T cells.

mitogen ConA. We used relatively low doses of ConA, since a reduced proliferative T cell response in Sdc1-deficient T cells was found in co-culture with unstimulated DC which in general are weak T cell stimulators. Indeed, stimulation by ConA (0.1–0.25 μg/ml) revealed a reduced proliferative response of Sdc-1 deficient splenocytes compared to WT splenocytes (Fig 3A). Furthermore, ConA induced IL-17 production was lower in Sdc1-deficient splenocytes compared to WT splenocytes, whereas IL-10 and INF-y production were similar (Fig 3B). Levels of IL-4, IL-21 and TNF-α were undetectable. An explanation for the difference in proliferative capacity of Sdc-1-deficient and WT splenocytes upon ConA activation could be a different composition of total splenocytes. However, Sdc-1-deficient and WT splenocyte displayed a similar distribution of lymphocyte subsets (S2 Fig). Another explanation for the

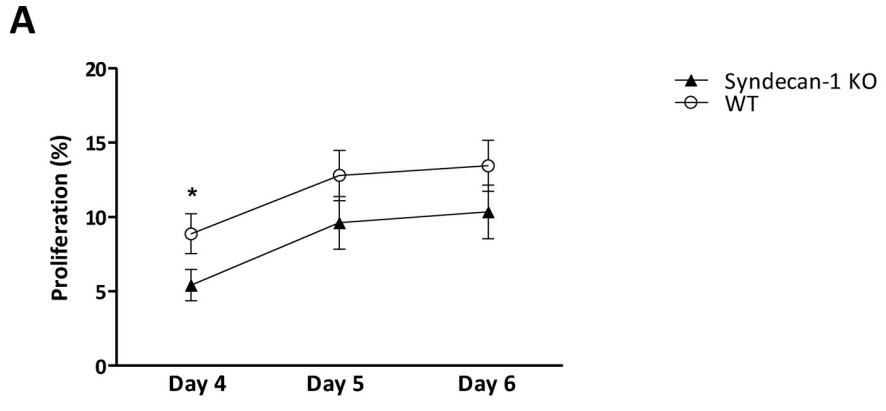

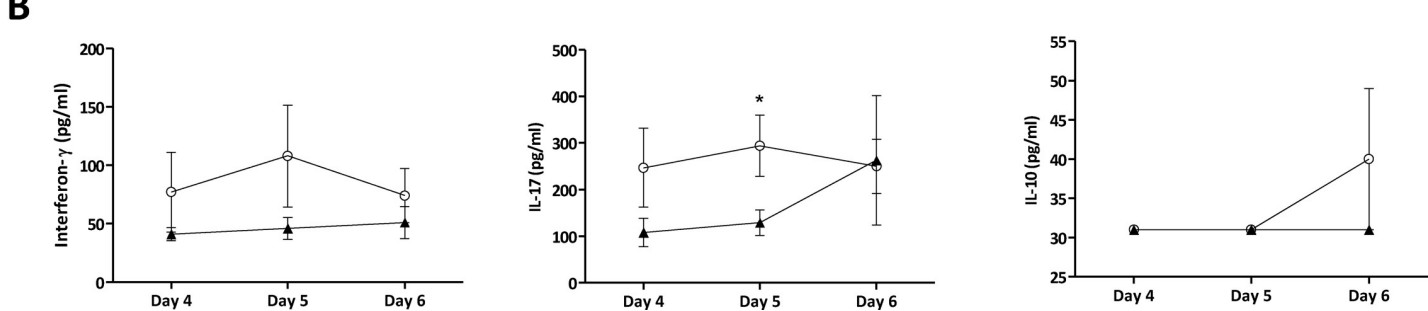

**Fig 3. Proliferation and cytokine production are reduced in Sdc-1 deficient splenocytes upon stimulation with ConA.** Proliferation of WT and Sdc-1 deficient splenocytes was determined in the presence of 0.1–0.25 μg/ml ConA as analyzed by CFSE dilution using flow cytometry (A). Interferon-γ, IL-17 and IL-10 production by WT and Sdc-1 deficient splenocytes was determined in the presence of 0.25 μg/ml ConA (B), as determined by ELISA. Proliferation and cytokine levels are expressed as means and standard error of means of 10 independent experiments. * p<0.05, Mann Whitney test, unpaired t-test.

observed differences in proliferative response of Sdc-1 and WT splenocytes could be a difference in cell viability. According to literature, Sdc-1 appears to affect cell viability, as Sdc-1 has been reported to prevent epithelial apoptosis and its absence in specific types of cancer results in cell cycle arrest and early apoptosis [40–44]. Nevertheless, incubation of Sdc-1 deficient splenocytes with low dose 4-nitroquinoline 1-oxide resulted in similar degrees of apoptosis compared to WT splenocytes (S3 Fig) and could not explain the observed difference in proliferative response.

In conclusion, Sdc-1 deficiency results in a reduced proliferative T cell response, both in co-cultures with DC as well as upon ConA stimulation.

## Sdc-1 deficiency does not affect allograft survival

The HS chains of Sdc-1 are involved in CXCL1 binding and function as a co-receptor for CXCR2, i.e. the receptor for CXCL1, thereby mediating activation of neutrophils [8, 9, 19]. Therefore, the absence of Sdc-1 in an allograft could reduce local CXCL1 concentrations, and as a consequence inflammation and allograft rejection. We determined the survival of Sdc-1 deficient versus WT allografts in a fully mismatched heterotopic heart transplant model, with Balb/c mice serving as recipient. Sdc-1 deficiency did not prolong allograft survival (median survival 10 days, n = 9) compared to WT allograft survival (median survival 11 days, n = 8), as shown in Fig 4A. In plasma of recipients of a Sdc-1 deficient allograft we found increased levels of CXCL1, while IL-4, IL-10, IL-17 and interferon-γ levels did not differ from recipients with a WT allograft (Fig 4B).

The reduced proliferative response of T cells observed in case of Sdc-1 deficiency might impair allogeneic responses. Furthermore, allograft rejection has been associated with increased IL-17 levels due to increased Th17 activity [45–47], and an early rise in IL-17 levels was shown to induce Th1 differentiation, thereby contributing to allograft rejection [48]. We found reduced IL-17 production by Sdc-1 deficient responder T cells stimulated by ConA (Fig 3). Therefore, we hypothesized that allograft survival could be prolonged in Sdc-1 deficient mice compared to WT recipients. However, Balb/c allograft survival in Sdc-1 deficient recipients was not prolonged (median survival 8.5 days, n = 8) compared to allograft survival in WT recipients (median survival 9.5 days, n = 9), as depicted in Fig 4C. IL-17 and IL-4 levels were undetectable in plasmas from both Sdc-1 deficient and WT recipients, while no differences were found in IL-10 levels. However, INF-γ and CXCL1 levels were higher in Sdc-1 deficient recipients compared to WT recipients (Fig 4D), which suggests an increased Th1 response in Sdc-1 deficient recipients.

In conclusion, in an allogeneic heart transplant model, Sdc-1 deficiency in either allograft or recipient did not affect allograft survival. The cytokine profile in plasma of Sdc-1 deficient recipient mice suggests an increased Th1 response compared to WT recipients.

## Discussion

Sdc-1 can contribute to immunological responses via binding of cytokines on the cell surface, removal of local cytokines by shedded Sdc-1, and functioning as (co-) receptor. Its immuno-modulatory properties have been shown in various mouse models [11–13, 15–17, 20, 49]. While the effects of Sdc-1 deficiency in these inflammatory models have been well investigated, there are few data on the specific role of Sdc-1 in the functioning of different immune cells and on the role of Sdc-1 in allograft rejection. In this study, we used Sdc-1 deficient mice to analyze the role of Sdc-1 in a transplantation model and we especially focused on the contribution of Sdc-1 to DC–T cell interaction. Taken together, we found no effect of Sdc-1 deficiency on the phenotype or T cell stimulating capacity of DC, but showed reduced T cell proliferation and

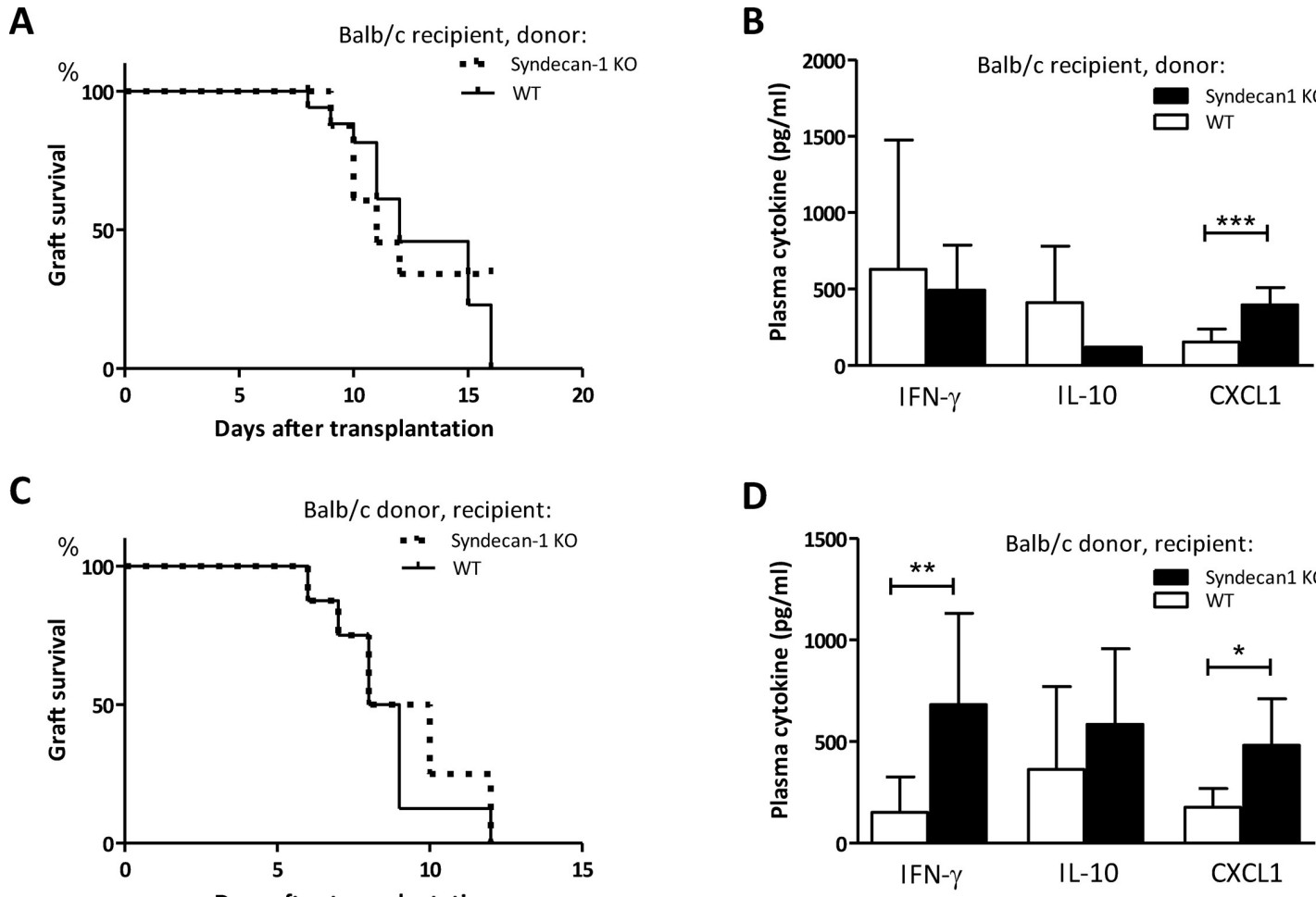

**Fig 4. Sdc-1 deficiency affects cytokine levels in allograft recipients without affecting allograft survival.** Allograft survival in a fully mismatched heterotopic heart transplantation model. Hearts were obtained from Sdc-1-deficient (n = 9) or WT mice (n = 8) and transplanted in Balb/c mice (A). Plasma cytokine levels in Balb/c recipient mice that received WT or Sdc-1 deficient hearts (B). Allograft survival of Balb/c hearts in WT (n = 8) or Sdc1-deficient (n = 8) recipients (C). Plasma cytokine levels in WT or Sdc-1 deficient recipients that received a Balb/c heart (D). * p< 0.05, ** p< 0.01, *** p< 0.001, Mann Whitney test.

reduced IL-17 production by Sdc-1-deficient T cells in co-cultures with DC or upon stimulation with ConA. This effect of Sdc-1 deficiency was especially observed when T cells were cocultured with unstimulated DC and relatively low doses of ConA. Although unstimulated DC are usually weak inducers of T cell proliferation, the activation state of these DC during coculture appeared sufficient to induce T cell proliferation in a fully MHC-mismatched stimulator responder combination, as observed before [33], which allowed to evaluate the effect of Sdc-1 deficiency. These manifestations of altered T cell function due to Sdc-1 deficiency were not reflected in delayed allograft rejection as allograft survival was not affected by Sdc-1 deficiency of graft recipients.

We could not demonstrate surface expression of Sdc-1 on T cells, but detected intracellular Sdc-1 expression in ConA activated T cells. Since Sdc-1 expression has been described in the nucleus of various cell types [43, 50–52], we suggest an intracellular role for Sdc-1 in T cell functioning. Other studies have shown that nuclear Sdc-1 can regulate gene transcription and cell proliferation [53, 54]. Further delineation of the subset of T cells expressing Sdc-1 as well as analysis of intracellular signaling cascades, resulting in IL-17 production or cell

proliferation, in Sdc-1 deficient and WT T cells could shed more light on the exact role of Sdc-1 in T cell functioning.

Acute cellular allograft rejection is a T cell-dependent phenomenon and has recently been related to a Th17 response reflected by increased IL-17 levels in acute rejection [45, 46, 55]. Based on our *in vitro* observation with reduced IL-17 production by Sdc-1-deficient T cells after stimulation with ConA, we postulated that Sdc-1 deficiency could be beneficial for allograft survival. Nevertheless, we found no change in allograft survival regardless of whether the recipient or the donor of the allograft was Sdc-1-deficient. It has to be noticed that we used a donor-recipient combination with a complete MHC mismatch, resulting in strong alloreactive response which could have obscured more subtle effects of Sdc-1 deficiency. We were also unable to detect an effect of Sdc-1 deficiency on IL-17 plasma levels, since IL-17 was undetectable in plasma of Sdc-1 deficient as well as WT graft recipients.

Although the role of Sdc-1 in allograft rejection was not investigated before, Sdc-1 deficiency was associated with increased inflammation and worse outcome in several animal models of inflammation [11–17]. Based on these data, one might expect an accelerated graft rejection in Sdc-1 deficient graft recipients. At the time of graft rejection, we found increased IFN-γ levels in plasmas of Sdc-1 deficient recipients as compared to WT recipients. This may indicate an enhanced Th1 response in case of Sdc-1 deficiency although this was not reflected in accelerated allograft rejection. In experimental autoimmune encephalomyelitis, which is a T cell mediated inflammatory model, Sdc-1 deficient mice showed enhanced disease severity compared to WT mice, with increased Th1 and Th17 infiltration in the brain [14]. In adoptive transfer experiments, this effect was shown to be independent of Sdc-1 deficiency of T cells. Moreover, in chimeric WT mice injected with Sdc-1 deficient bone marrow, lymph node cells and splenocytes harvested at the peak of disease severity showed increased IFN-γ production upon antigen specific re-stimulation *in vitro* as compared to Sdc-1 deficient mice injected with WT bone marrow. These and our results suggest systemic immunological effects, beyond T cell functioning, of Sdc-1 deficiency.

Besides an increase in interferon-γ, we also found elevated plasma levels of CXCL1 in Sdc-1 deficient recipients at time of allograft rejection. Since Sdc-1 binds CXCL1 via its HS-chains, and bound CXCL1 can be removed by Sdc-1 shedding, higher circulating CXCL1 levels can be expected in the absence of Sdc-1.

In conclusion, we found no role for Sdc-1 in DC but a clear role of Sdc-1 in T cell functioning, as reflected by reduced proliferation of Sdc-1 deficient T cells upon stimulation by DC or ConA. Based on our results and literature data we suggest a role for intracellular Sdc-1 in T cell functioning. In a strongly mismatched heart transplant model, these effects on T cell functioning did not translate into a prolongation of allograft survival.

## Supporting information

**S1 Fig. Sdc-1 deficiency does not affect TLR2 or TR9 triggered DC maturation or function.** Expression of co-stimulatory molecules and MHCII by TLR2 (1 μg/ml PAM$_3$CysSerLys$_4$) and TLR9 (5 μg/ml ODN1826) stimulated WT and Sdc-1-deficent CD11c$^+$ DC was measured by flow cytometry (A). Stimulation with ODN1826 resulted in decreased CD80 expression in Sdc-1 KO as compared to WT DC. Cytokine profile of WT and Sdc-1-deficent DC as measured by ELISA (B). T cell stimulatory capacity of TLR2 or TLR9 stimulated WT and Sdc-1-deficient DC as measured by CFSE dilution using flow cytometry (C). Experiments were replicated 3–5 times. Results are expresses as mean ± standard error of means. $^*$ p< 0.05 (Mann Whitney test).
(TIF)

**S2 Fig. Splenocyte composition is not altered in Sdc-1 deficient mice.** Splenocytes of Sdc-1 deficient and WT mice were analyzed for expression of CD3, CD4, CD8 and CD19 by flow cytometry. Experiments were replicated 5 times. Results are expresses as mean ± standard error of means.
(TIF)

**S3 Fig. Sdc-1 splenocytes are not more susceptible to 4-nitroquinoline 1-oxide induced apoptosis.** Sdc-1 deficient or WT splenocytes were incubated with low dose 4-nitroquinoline 1-oxide, stained with Annexin V–propidium iodide and analyzed by flow cytometry. Experiments were replicated 3 times. Results are expresses as mean ± standard error of means.
(TIF)

## Author Contributions

**Conceptualization:** A. Rops, L. Hilbrands, J. van der Vlag.

**Data curation:** M. Kouwenberg, M. Bakker-van Bebber, L. Diepeveen.

**Formal analysis:** M. Kouwenberg.

**Funding acquisition:** A. Rops, L. Hilbrands, J. van der Vlag.

**Investigation:** M. Kouwenberg.

**Methodology:** M. Kouwenberg, L. Hilbrands.

**Project administration:** M. Kouwenberg.

**Resources:** M. Götte.

**Software:** M. Kouwenberg.

**Supervision:** A. Rops, L. Hilbrands, J. van der Vlag.

**Visualization:** M. Kouwenberg.

**Writing – original draft:** M. Kouwenberg.

**Writing – review & editing:** M. Kouwenberg, A. Rops, M. Götte, L. Hilbrands, J. van der Vlag.

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
