## [Decision Letter · Decision Letter 0]

30 Apr 2020

PONE-D-20-06732

Role of syndecan-1 in the interaction between dendritic cells and T cells

PLOS ONE

Dear Dr. Vlag,

Thank you for submitting your manuscript to PLOS ONE. After careful consideration, we feel that it has merit but does not fully meet PLOS ONE’s publication criteria as it currently stands. Therefore, we invite you to submit a revised version of the manuscript that addresses the points raised during the review process. Please consider and address each of the comments raised by the reviewers before resubmitting the manuscript.

We would appreciate receiving your revised manuscript by Jun 14 2020 11:59PM. To enhance the reproducibility of your results, we recommend that if applicable you deposit your laboratory protocols in protocols.io, where a protocol can be assigned its own identifier (DOI) such that it can be cited independently in the future. For instructions see: http://journals.plos.org/plosone/s/submission-guidelines#loc-laboratory-protocols

We look forward to receiving your revised manuscript.

Kind regards,

Senthilnathan Palaniyandi, Ph.D

Academic Editor

PLOS ONE

Journal Requirements:

'This study was financially supported by NWO ZonMw AGIKO 92003567 Radboud PhD program 2010,

and Institute for Infection, Inflammation & Immunity (N4i), Radboud University Nijmegen Medical

Centre, Nijmegen, The Netherlands.'

'JvdV; LH; MK: NWO ZonMw AGIKO. 92003567, Radboud PhD programm 2010. The funders had no role in study design, data collection and analysis, decision to publish, or preparation of the manuscript.'

Additional Editor Comments (if provided):

Reviewers' comments:

Reviewer's Responses to Questions

**Comments to the Author**

1. Is the manuscript technically sound, and do the data support the conclusions?

Reviewer #1: Yes

Reviewer #2: No

2. Has the statistical analysis been performed appropriately and rigorously? 

Reviewer #1: Yes

Reviewer #2: No

3. Have the authors made all data underlying the findings in their manuscript fully available?

Reviewer #1: Yes

Reviewer #2: No

4. Is the manuscript presented in an intelligible fashion and written in standard English?

Reviewer #1: Yes

Reviewer #2: No

5. Review Comments to the Author

Reviewer #1: Kouwenberg M et al., studied the interaction between dendritic cells (DC) and T- cells deciphering the role of Syndecan-1 in the context of graft survival in an allogeneic transplant model. The authors found that Syndecan-1 depletion did not affect cytokine production, expression of co-stimulatory molecules, or T cell stimulatory capacity of DC. Additionally, ConA activation of T-cells by ConA or DC induced Syndecan-1 expression associated with lower proliferative potential and IL-17 production. This study suggests a role for intracellular Sdc-1 in function of T cells, however, without beneficial effect for prolongation of allograft survival with a complete MHC mismatch. Overall, the study is well-designed and well-written. Only one note; a recent article should be mentioned as they are among the few that have shown the correlation between Syndecan-1 and T-cells (Saleh ME, et al., The Immunomodulatory Role of Tumor Syndecan-1 (CD138) on Ex Vivo Tumor Microenvironmental CD4+ T Cell Polarization in Inflammatory and Non-Inflammatory Breast Cancer Patients, PLoS One 2019 May 30;14(5):e0217550).

Reviewer #2: The study by Kouwenberg et al.is aimed at investigating the “Role of syndecan-1 in modulating the outcomes of the interactions between dendritic cells and T cells”. Although the goal of the study is interesting, the manuscript is severely lacking. The study is not well designed and poorly executed and the results over-interpreted. Hence, it is not suitable for publication in Plos One.

Here is a summary of the major concerns.

Major comments:

Introduction:

The authors left highly relevant manuscripts (PMID: 26300525, 30045969) about the expression of sdc1 and function sdc1 on NKT and γδ T cells out of their introduction and consequently the subject and status of the field are poorly presented to the reader.

Methods section:

The methods were poorly described with severely limited key assays, including MLR, apoptosis assay, antibodies used (left blank), and ELISA.

Results:

Fig. 1.

- Fig. 1B-right graph. Although there was no apparent significant differences in the expression of CD40 between LPS stimulated Wt and sdc1 KO DCs, the authors show significant difference, whereas the opposite appears to be true for MHC class II where there an apparent differences but no statistically is noted by the authors.

- Figs.1D-E. The authors did not show how T cells were isolated? Need better explanation about purity, numbers and viability and composition of purified T cells.

Fig. 2:

1. It is unclear how MLR assay was performed. The experiment was not properly designed and doesn’t have a proper negative control (no lymphocyte culture only).

2. Staining for sdc1 is not convincing, the signal is very weak. Other studies did not detect sdc1 on T cells, but only on NKT and γδ T cells (PMID: 26300525, 30045969). These studies should be cited and discussed.

3. How do the authors explain activation of T cells with unstimulated DCs and how sdc1 affected the outcome? These pointes should have been discussed.

Fig. 3

1. The data in Fig.3 are the only interesting observations showing proliferation of Sdc-1-deficient T cells was significantly reduced as compared to WT T cells including after ConA stimulation. However, it is unclear what cell type(s) among splenocytes is/are dividing, given splenocytes are mixtures of conventional T cells, NKT, γδ T, B cells, macrophages and DCs. Nowhere in the manuscript do the authors tell us what T cell type (CD4, CD8, NKT cells) express sdc1?

It is important to have representative flowcytometric plots to accompany each graph so the readers can evaluate the data.

Minor comments,

Line 240, The authors said “Cell surface expression of Sdc-1 has been described for human T cells (35, 36), but not for mouse T cells” which is not true. There are several studies that examined the role of Sdc-1 in mouse T cells. (PMID: 26300525, 30045969).

The manuscript should be fully proofread for typos and grammars.

6. PLOS authors have the option to publish the peer review history of their article (what does this mean?). If published, this will include your full peer review and any attached files.

Reviewer #1: No

Reviewer #2: No

---

## [Author Response · Author response to Decision Letter 0]

8 Jun 2020

Reviewer #1

We thank the reviewer for the very positive remarks.

“A recent article should be mentioned as they are among the few that have shown the correlation between Syndecan-1 and T-cells (Saleh ME, et al., The Immunomodulatory Role of Tumor Syndecan-1 (CD138) on Ex Vivo Tumor Microenvironmental CD4+ T Cell Polarization in Inflammatory and Non-Inflammatory Breast Cancer Patients, PLoS One 2019 May 30;14(5):e0217550)”

This paper has been added to the references and the main findings are included in the introduction section (lines 74 to 75).

Reviewer #2

We thank the reviewer for the critical comments. This enabled us to further improve the manuscript considerably.

“The authors left highly relevant manuscripts (PMID: 26300525, 30045969) about the expression of sdc1 and function sdc1 on NKT and γδ T cells out of their introduction and consequently the subject and status of the field are poorly presented to the reader.”

We thank the reviewer for drawing our attention to these two relevant papers. The articles have been added to the references and main findings of these papers are included in the introduction (lines 68 to 69, 86 to 87) and results (line 252 to 253) sections.

The methods were poorly described with severely limited key assays, including MLR, apoptosis assay, antibodies used (left blank), and ELISA

In our Methods and Material section we have added details to the description of the MLR, proliferation assay and apoptosis assay (lines 139 to 148). For an extensive description of the apoptosis assay, we referred to a previous publication (1). 

Fig. 1.

- Fig. 1B-right graph. Although there was no apparent significant differences in the expression of CD40 between LPS stimulated Wt and sdc1 KO DCs, the authors show significant difference, whereas the opposite appears to be true for MHC class II where there an apparent differences but no statistically is noted by the authors.

We think that we have caused a misunderstanding and apologize for that. The asteriks in figure 1 indicate a statistically significant difference for the comparison between the unstimulated and LPS matures DC. Indeed, there was no significant upregulation of MHC -II in WT DC upon LPS due to a large standard deviation (P=0.095).

We have clarified this in the legend of Fig. 1B

- Figs.1D-E. The authors did not show how T cells were isolated? Need better explanation about purity, numbers and viability and composition of purified T cells.

As described in our Methods and Material section, T cells were enriched from splenocytes by depleting MHC-II positive cells with magnetic cell sorting (lines 130 to 132). We have added details on purity of the resulting cell population (line 132 to 133). Based on vigorous cell proliferation in our positive control condition (stimulation with 2 µg/ml concanavalinA) we have no doubts on cell viability after T cell enrichment and CFDA-SE staining.

Fig. 2:

1. It is unclear how MLR assay was performed. The experiment was not properly designed and doesn’t have a proper negative control (no lymphocyte culture only).

We apologize for not being clear. The current description of the MLR assay includes all details (lines 139 to 142). We included a negative control (responder T cells only) in all experiments, and observed negligible proliferation in this condition (max. 2-4% at day 6).

2. Staining for sdc1 is not convincing, the signal is very weak. Other studies did not detect sdc1 on T cells, but only on NKT and γδ T cells (PMID: 26300525, 30045969). These studies should be cited and discussed.

As mentioned above, these studies have been cited in the manuscript in the sections introduction (lines 68 to 69, 86 to 87) and results (line 252 to 253) sections.

3. How do the authors explain activation of T cells with unstimulated DCs and how sdc1 affected the outcome? These pointes should have been discussed.

We agree with the reviewer that proliferation of T cells in coculture with unstimulated DCs is expected to be low. Notably, MLR was performed with responder and stimulator cells from mice of different haplotypes (C57Bl/6 H-2d and Balb/c H-2b, respectively), resulting in a full MHC-mismatch. Although expression of co-stimulatory molecules is low in unstimulated DC at the start of the coculture, the expression of co-stimulatory molecules may increase during coculture, enhancing the stimulatory capacity of these cells. This is in agreement with similar observations in a previous study (2). We have added these considerations to the discussion section (lines 351 to 356).

In the discussion section we also speculate about the role of intracellular Sdc-1 in explaining our findings of reduced proliferation of Sdc-1 deficient T cells. 

Fig. 3

1. The data in Fig.3 are the only interesting observations showing proliferation of Sdc-1-deficient T cells was significantly reduced as compared to WT T cells including after ConA stimulation. However, it is unclear what cell type(s) among splenocytes is/are dividing, given splenocytes are mixtures of conventional T cells, NKT, γδ T, B cells, macrophages and DCs. Nowhere in the manuscript do the authors tell us what T cell type (CD4, CD8, NKT cells) express sdc1?

It is important to have representative flowcytometric plots to accompany each graph so the readers can evaluate the data.

In Figure 3 we show a decreased proliferative response of Sdc-1 deficient splenocytes as compared to WT splenocytes. We have analyzed splenocyte composition by flow cytometry (CD3, CD4, CD8 and CD19) as shown in supplementary figure 2. We did not include analysis of NKT cells, since these cells are usually less prevalent in the spleen (1-2%). Similarly low percentages have been reported for gamma-delta T cells. Importantly, the splenocyte composition was not affected by Sdc-1-deficiency.

Syndecan-1 expression was detected solely intracellular in ConA activated splenic T cells (CD3-positive). ConA is a known T cell mitogen that does not induce B cell proliferation (3, 4). Liver NKT cell proliferation has been described upon non-hepatotoxic infusion of Con-A in mice (5). However, our finding of an increase in intracellular expression of Sdc-1 is unlikely to reflect NKT proliferation, since the described NKT cell proliferation by Miyagi et al. concerned the NKT1 subset (as reflected by increased interferon-� levels), and these cells do not express Sdc-1. As emphasized in the discussion, further research should focus on the exact T cell subsets expressing Sdc-1 and which mechanisms underly Sdc-1-mediated T cell proliferation and IL-17 production (lines 363 to 365).

Line 240, The authors said “Cell surface expression of Sdc-1 has been described for human T cells (35, 36), but not for mouse T cells” which is not true. There are several studies that examined the role of Sdc-1 in mouse T cells. (PMID: 26300525, 30045969).

We have corrected this in the revised manuscript. 

References

1. Dieker J, Hilbrands L, Thielen A, Dijkman H, Berden JH, van der Vlag J. Enhanced activation of dendritic cells by autologous apoptotic microvesicles in MRL/lpr mice. Arthritis research & therapy. 2015;17:103.

2. Kouwenberg M, Jacobs CW, van der Vlag J, Hilbrands LB. Allostimulatory Effects of Dendritic Cells with Characteristic Features of a Regulatory Phenotype. PloS one. 2016;11(8):e0159986.

3. Hawrylowicz CM, Klaus GG. Activation and proliferation signals in mouse B cells. IV. Concanavalin A stimulates B cells to leave G0, but not to proliferate. Immunology. 1984;53(4):703-11.

4. Piguet PF, Dewey HK, Vassalli P. Induction or suppression of B cell proliferation and differentiation by phytohemagglutinin or concanavalin A in mouse spleen cell cultures. Journal of immunology (Baltimore, Md : 1950). 1976;117(5 Pt.2):1817-23.

5. Miyagi T, Takehara T, Tatsumi T, Suzuki T, Jinushi M, Kanazawa Y, et al. Concanavalin a injection activates intrahepatic innate immune cells to provoke an antitumor effect in murine liver. Hepatology. 2004;40(5):1190-6.

---

## [Decision Letter · Decision Letter 1]

6 Jul 2020

Role of syndecan-1 in the interaction between dendritic cells and T cells

PONE-D-20-06732R1

Dear Dr. Vlag,

We’re pleased to inform you that your manuscript has been judged scientifically suitable for publication and will be formally accepted for publication once it meets all outstanding technical requirements.

Kind regards,

Senthilnathan Palaniyandi, Ph.D

Academic Editor

PLOS ONE

Additional Editor Comments (optional):

Reviewers' comments:

Reviewer's Responses to Questions

**Comments to the Author**

1. If the authors have adequately addressed your comments raised in a previous round of review and you feel that this manuscript is now acceptable for publication, you may indicate that here to bypass the “Comments to the Author” section, enter your conflict of interest statement in the “Confidential to Editor” section, and submit your "Accept" recommendation.

Reviewer #1: All comments have been addressed

Reviewer #2: All comments have been addressed

2. Is the manuscript technically sound, and do the data support the conclusions?

Reviewer #1: Yes

Reviewer #2: Yes

3. Has the statistical analysis been performed appropriately and rigorously? 

Reviewer #1: Yes

Reviewer #2: Yes

4. Have the authors made all data underlying the findings in their manuscript fully available?

Reviewer #1: Yes

Reviewer #2: Yes

5. Is the manuscript presented in an intelligible fashion and written in standard English?

Reviewer #1: Yes

Reviewer #2: Yes

6. Review Comments to the Author

Reviewer #1: The authors of the manuscript ''Role of syndecan-1 in the interaction between dendritic cells and T cells'' have improves significantly the revised manuscript. I recommend acceptance in its present form

Reviewer #2: (No Response)

7. PLOS authors have the option to publish the peer review history of their article (what does this mean?). If published, this will include your full peer review and any attached files.

Reviewer #1: No

Reviewer #2: No

---

## [Editor Report · Acceptance letter]

9 Jul 2020

PONE-D-20-06732R1 

Role of syndecan-1 in the interaction between dendritic cells and T cells 

Dear Dr. van der Vlag:

I'm pleased to inform you that your manuscript has been deemed suitable for publication in PLOS ONE. Congratulations! Your manuscript is now with our production department. 

Kind regards, 

on behalf of

Dr. Senthilnathan Palaniyandi 

Academic Editor

PLOS ONE